# Increased Density of Growth Differentiation Factor-15+ Immunoreactive M1/M2 Macrophages in Prostate Cancer of Different Gleason Scores Compared with Benign Prostate Hyperplasia

**DOI:** 10.3390/cancers14194591

**Published:** 2022-09-22

**Authors:** Gabriel A. Bonaterra, Alexander Schleper, Maximilian Skowronek, Lucia S. Kilian, Theresa Rink, Hans Schwarzbach, Hendrik Heers, Jörg Hänze, Peter Rexin, Annette Ramaswamy, Carsten Denkert, Beate Wilhelm, Axel Hegele, Rainer Hofmann, Eberhard Weihe, Ralf Kinscherf

**Affiliations:** 1Department of Medical Cell Biology, Institute for Anatomy and Cell Biology, University of Marburg, 35037 Marburg, Germany; 2Department of Urology and Pediatric Urology, University Medical Center Marburg, 35043 Marburg, Germany; 3Institute of Pathology Aurich, 26603 Aurich, Germany; 4Institute of Pathology, UKGM University Hospital Marburg, 35043 Marburg, Germany; 5Department of Radiotherapy and Radiooncology, UKGM University Hospital Marburg, 35043 Marburg, Germany; 6Urological Center Mittelhessen/DRK-Hospital Biedenkopf, 35216 Biedenkopf, Germany

**Keywords:** benign prostatic hyperplasia, biomarker, GDF-15, Gleason scores, innervation, lymphocytes, macrophages, prostate cancer

## Abstract

**Simple Summary:**

Prostate cancer (PCa) is the second most diagnosed cancer and cause of death in men worldwide. The main challenge is to discover biomarkers for malignancy to guide the physician towards optimized diagnosis and therapy. There is recent evidence that growth differentiation factor-15 (GDF-15) is elevated in cancer patients. Therefore, we aimed to decipher GDF-15+ cell types and their density in biopsies of human PCa patients with Gleason score (GS)6–9 and benign prostate hyperplasia (BPH). Here we show that the density of GDF-15+ cells, mainly identified as interstitial macrophages (MΦ), was higher in GS6–9 than in BPH, and, thus, GDF-15 is intended to differentiate patients with high GS vs. BPH, as well as GS6 vs. GS7 (or even with higher malignancy). Some GDF-15+ MΦ showed a transepithelial migration into the glandular lumen and, thus, might be used for measurement in urine/semen. Taken together, GDF-15 is proposed as a novel tool to diagnose PCa vs. BPH or malignancy (GS6 vs. higher GS) and as a potential target for anti-tumor therapy. GDF-15 in seminal plasma and/or urine could be utilized as a non-invasive biomarker of PCa as compared to BPH.

**Abstract:**

Although growth differentiation factor-15 (GDF-15) is highly expressed in PCa, its role in the development and progression of PCa is unclear. The present study aims to determine the density of GDF-15+ cells and immune cells (M1-/M2 macrophages [MΦ], lymphocytes) in PCa of different Gleason scores (GS) compared to BPH. Immunohistochemistry and double immunofluorescence were performed on paraffin-embedded human PCa and BPH biopsies with antibodies directed against GDF-15, CD68 (M1 MΦ), CD163 (M2 MΦ), CD4, CD8, CD19 (T /B lymphocytes), or PD-L1. PGP9.5 served as a marker for innervation and neuroendocrine cells. GDF-15+ cell density was higher in all GS than in BPH. CD68+ MΦ density in GS9 and CD163+ MΦ exceeded that in BPH. GDF-15+ cell density correlated significantly positively with CD68+ or CD163+ MΦ density in extratumoral areas. Double immunoreactive GDF-15+/CD68+ cells were found as transepithelial migrating MΦ. Stromal CD68+ MΦ lacked GDF-15+. The area of PGP9.5+ innervation was higher in GS9 than in BPH. PGP9.5+ cells, occasionally copositive for GDF-15+, also occurred in the glandular epithelium. In GS6, but not in BPH, GDF-15+, PD-L1+, and CD68+ cells were found in epithelium within luminal excrescences. The degree of extra-/intra-tumoral GDF-15 increases in M1/M2Φ is proposed to be useful to stratify progredient malignancy of PCa. GDF-15 is a potential target for anti-tumor therapy.

## 1. Introduction

Prostate cancer (PCa) is the second most frequently diagnosed cancer and the second cause of death in men among most developed countries; its incidence has increased significantly over recent years [1]. The clinical outcomes for patients with a 5-year survival rate is only 30% [1]. The main challenge of PCa is to discover adequate severity indicators (so far insufficient) to guide the physician towards an accurate diagnosis of the degree of malignancy and to select the most appropriate therapy [2]. The histopathological diagnosis of PCa first requires light microscopic examination of hematoxylin and eosin (HE)-stained biopsies to deliver the histological diagnosis of adenocarcinoma of the prostate by the evaluation of three main criteria: glandular morphology, losing of basal epithelial cells, and nuclear and/or nucleolar enlargement [3]. Many of the common growth patterns of prostatic adenocarcinoma are well illustrated by the International Society of Urological Pathology (ISUP) based on modified Gleason grading/scores (GS) [4].

In general, cancer is defined by the intrinsic activity of tumor cells, the grade of angiogenesis, and the immune cells recruited into the tumor microenvironment (TME) [5]. The role of immune cells during the development of cancer, particularly in PCa, remains largely unclear. However, infiltration of immune cells in PCa has been reported to be positively associated with clinical outcomes [6]. Most recently, it has been suggested that tumor-infiltrating leukocytes (TILs) like B, T, and dendritic cells, monocytes/macrophages (MΦ), and neutrophils might control malignant growth [7]. In this context, TILs—as an important component of the TME—can affect the progression of breast [5], colorectal [6], ovarian [8], renal [9], and prostate cancers [10].

The differential diagnosis of high-grade adenocarcinoma of different origins is complex. New and more accurate tumor markers are needed to predict the aggressiveness and metastatic potential of a given carcinoma. In this context, a divergent member of the TGF-beta superfamily, i.e., growth differentiation factor-15 (GDF-15)—also known as MΦ inhibitory cytokine-1 (MIC-1), NSAID-activated gene-1, placental bone morphogenetic protein (PLAB) [11,12]—is overexpressed by a majority of cancers [13]. However, the role of GDF-15 in cancer development and progression depends on the cancer type, stage, and tumor microenvironment [14]. In vitro, GDF-15 is induced in many cell types, including neurons, in response to stress [15]. In vivo, the situation can be variable depending on the cellular environment that defines the stimulation or inhibition of GDF-15 in response to local stress [16]. Constitutive GDF-15 gene expression is relatively low in the non-diseased prostate, liver, kidney, pancreas, and fetal brain but abundant in the placenta [17]. GDF-15 can have opposite effects in different cancer cell lines, and has been shown to inhibit cell growth, to activate apoptosis, and to increase cancer invasiveness [18]. GDF-15 gene expression has been found to be three times higher in androgen-sensitive human prostate adenocarcinoma (LNCaP) cells compared to androgen-negative PC-3 cells [19]. Moreover, it has been shown that PCa cells induced cancer-associated fibroblasts and osteocytes to produce GDF-15, resulting in activation of cell proliferation, migration, and metastatic invasion; this may explain why the most frequent metastatic site for PCa cells is bone [19]. However, GDF-15 can also inhibit metastasis in p53-null human PCa cell lines [19]. In regard to its obvious pleiotropic and sometimes opposite effects, GDF-15 is suggested as a biomarker of tumor severity or as a diagnostic/therapeutic target against metastasis. However, controversial results from different studies showing elevated [20,21] or even reduced [22,23] expression of GDF-15/MIC-1 in PCa serum samples, create ambiguity regarding the expression pattern of GDF-15/MIC-1 with respect to PCa. The abundance of serum GDF-15 levels was associated with weight loss and cachexia in patients with advanced PCa [14,24]. GDF-15 has recently been proposed as a potential marker to discriminate between PCa and benign prostate hyperplasia (BPH) [25]. Additionally, GDF-15 is a novel regulator of PD-L1 expression in glioblastoma multiforme (GBM); thus, targeting the GDF15/PD-L1 pathway might be a promising treatment for GBM patients [26]. PD-L1 is expressed in immune, as well as tumor cells, being associated with the response to anti-PD-L1 immunotherapy [27]. PD-L1 expression correlates with response to immunotherapy in certain tumor types, and neutralization of GDF-15 activity could be a possible treatment to extend the benefits of immunotherapy in patients with solid cancers and metastases [25,28].

The three most common cell types of prostatic epithelium following secretory and basal cells are neuroendocrine (NE) cells [29,30], which are part of the so-called diffuse system of amine precursor uptake and decarboxylation (APUD) [31]. NE cells display epithelial, endocrine, as well as neuronal characteristics, with nerve-like dendritic branching [32]. Frequently, PCa contain scattered NE cells [33] that mostly resemble other PCa cells in light microscopy [32]. NE malignant cells can be detected specifically by immunohistochemistry using markers such as chromogranin A or PGP 9.5 [34,35]. Increasing evidence suggests that the nervous system participates in all stages of cancer development [36,37,38]. In this context, nerve fibers and neurons have been recognized as essential components of the TME that favor the initiation and progression of a variety of solid tumors, as well as prostate intraepithelial neoplasia (PIN) [36,37,38]. In PCa, the stimulatory effect of tumor innervation and neurosignaling seems to be initiated by neurotrophic growth factors [39]. Clinical and epidemiological evidence show reduced incidence of PCa in patients with spinal cord injuries or patients treated with β-blockers, suggesting innervation-dependent PCa development [39,40].

The main objective of this study was to decipher specific signatures of GDF-15 immunoreactive (IR) cells in PCa of GS6–9 in relation to innervation, infiltration of immune cells (B and T cells, M1-, M2- MΦ) as compared to BPH.

## 2. Materials and Methods

### 2.1. Tissue Specimens and (Immuno) Histochemistry (IHC)

All procedures in this study were performed in accordance with the ethical standard of the institution, and the German law on the donation, removal, and transfer of organs (http://www.bgbl.de/xaver/bgbl/start.xav?startbk=Bundesanzeiger_BGBl&jumpTo=bgbl197s2631.pdf, accessed on 13 June 2022). The donors gave implicit consent to make use of the samples for research according to European regulations and in accordance with the Declaration of Helsinki, 1964 and its later amendments (https://www.wma.net/policies-post/wma-declaration-of-helsinki-ethical-principles-for-medical-research-involving-human-subjects/5/, accessed on 13 June 2022). Additionally, the use and examination of the PCa (GS6-GS9) and BPH biopsies were approved by the ethics committee of the medical faculty of the Philipps University of Marburg (AZ: Study 95/15). The patients’ personal information was subject to medical ethics confidentiality and protected against access by third parties. All specimens of PCa with different GS (GS6, *n* = 6; GS7, *n* = 6; GS8, *n* = 4; GS9, *n* = 6) and BPH (*n* = 4), were obtained from urology patients at the University Hospital of Giessen and Marburg after radical prostatectomy. The samples were fixed in buffered 4% formalin and paraffin-embedded. After fixation, the samples were washed with PBS and dehydrated by an ascending series of alcohols (50%, 70%, 90%, 100%, isopropanol). The material was then transferred to hot paraffin (>60 °C; 5–6 h) and after hardening, paraffin-embedded tissue was subsequently cut using a microtome. Sections were dewaxed for single and double immunohistochemical staining and conventional histochemical techniques. HE staining was conducted and the histopathological classification of the PCa according to Gleason [4] and the differential diagnosis of BPH were carried out by specialists of the Institute for Pathology at the UKGM Marburg, who also defined intratumoral (IT) and extratumoral (ET) regions of every single biopsy.. Antigen retrieval was conducted using sodium citrate and microwave at 600 W (2 min) and 360 W (10 min) or using proteolytic digestion with Pepsin/0.01 M HCl (0.4%) at room temperature (20 min). Immunoreactions were achieved by using the antibodies as described in Appendix A.

Single staining was performed after incubation of the sections with the primary antibody, and thereafter, detection was obtained with biotinylated rabbit anti-mouse (Dianova GmbH, Hamburg, Germany) or directly conjugated horseradish peroxidase (HRP)- or alkaline phosphatase (AP) antibodies (Linaris GmbH, Mannheim, Germany); using Vectastain ABC-Kit (Vector Laboratories Inc., Burlingame, CA, USA) (Appendix A). Finally, 3,3′-diaminobenzidine (DAB, Merck/Sigma-Aldrich Chemie GmbH, Munich, Germany) was used as detection system.

### 2.2. Double Immunofluorescence

After sodium citrate heat-induced or enzymatic antigen retrieval, the cross sections were incubated with two primary antibodies, using dilutions established by preliminary titration. Pairs of primary antibodies for double immunostaining were from different species (Appendix A). Detection was performed with donkey anti-rabbit biotin and streptavidin-Alexa Fluor™-488 (Invitrogen, Karlsruhe, Germany) or with donkey anti-rat IgG (H + L)-Cy3 (Dianova, Hamburg, Germany) (Appendix A). Nuclei were identified by DAPI (Sigma-Aldrich Chemie GmbH, Taufkirchen, Germany) staining (1 mg/mL).

### 2.3. Statistical Analyses

All statistical analyses were performed by using SigmaPlot 12^®^ (Systat Software Inc., San José, CA, USA). Statistical significance was determined by unpaired 2-tailed Student’s t-test. Normality test and equal variance test (Shapiro–Wilk and Brown–Forsythe) were performed. The U rank-sum W test (Mann–Whitney) was applied when data failed the normality and/or equal variance test. When appropriate, statistical significance was determined by one-way analysis of variance (ANOVA). Results are presented as means + standard error of the mean (SEM). *P* values of less than 0.05 (*p* ≤ 0.05) were considered as statistically significant. Data correlation was performed by using Pearson’s coefficient analysis.

## 3. Results

### 3.1. The Changed PGP9.5+-Innervation Matrix in PCa of Different GS as Compared to BPH

Immunoreactivity (IR) for PGP9.5 was found in nerve fibers and NE cells in PCa of all GS, as well as in the BPH. PGP9.5+ occurred as nerve bundles or partially elongated nerve fibers (Figure 1A). Most frequently, PGP9.5+ nerve fibers were localized in the stroma (Figure 1A); PGP9.5+ NE cells were found in the glandular epithelium (Figure 1B). The total PGP9.5+ area (ET and IT) in PCa of GS9 was significantly (*p* ≤ 0.05) 2.9-fold higher than in BPH, and 1.6-fold (*p* ≤ 0.05) higher than in PCa of GS6 (Figure 1B). The mean of the PGP9.5+ nerve fiber area in BPH was 0.15%, whereas the PGP9.5+ nerve fiber area in ET- and IT regions of all PCa samples varied between 0.28% (GS6) and 0.44% (GS9), and thus, was two- to three-fold (partly significantly) higher than in BPH (Figure 1B).

### 3.2. Increased Density of GDF-15+ Cells in PCa of Different GS as Compared to BPH

We determined the distribution and density of GDF-15+ cells in PCa of different GS (6–9) and in BPH as well. The majority of GDF-15+ cells were localized in the glands’ periphery, i.e., in the stroma (Figure 2A). In PCa of different GS, the density of GDF-15+ cells in ET and IT regions together, were significantly 51-fold (*p* ≤ 0.05) [GS6], 110-fold (*p* ≤ 0.01) [GS7], 129-fold [GS8], and 125-fold (*p* ≤ 0.05) [GS9] higher than in BPH (Figure 2B). Additionally, the density of GDF-15+ cells in PCa of GS7 was significantly (*p* ≤ 0.05) 2.2-fold higher than in GS6 (Figure 2B).

### 3.3. Increased Density of CD68+ and CD163+ Macrophages in PCa of Different GS as Compared to BPH

Because tumor-associated MΦ are suggested to be involved in tumor progression and/or aggressiveness, we investigated the distribution and density of CD68 (M1, pro-inflammatory) MΦ and CD163 (M2, anti-inflammatory) MΦ. CD68 IHC of BPH and PCa of GS6–9 revealed the typical morphological characteristics of MΦ (Figure 3A). Importantly, we found that MΦ form clusters in the ET and IT parenchyma, as well as in the glandular lumen (Figure 3A). Furthermore, isolated, multinucleated giant MΦ are seen in the ET and IT stroma and glandular lumen (Figure 3A). The density of CD68+ MΦ in PCa of GS9 was significantly (*p* ≤ 0.05) 4.9-fold higher than in BPH; additionally, increases—however, insignificant—were found in GS6 (2.3-fold), GS7 (3.2-fold), and GS8 (3.5-fold) in comparison to BPH (Figure 3B). In the ET regions of the PCa, the density of CD68+ MΦ correlated significantly (*p* ≤ 0.02) positively (*r* = 0.5) with the density of GDF-15+ cells in the ET regions (Figure 3C,D). However, densities of ET or IT CD68+ MΦ did not significantly correlate with the density of IT GDF-15+ cells (Figure 3C,D). The densities of CD68+, CD163+, and GDF15+ cells in IT regions compared with those in ET areas were insignificantly different.

CD163 is a well-known phenotypic marker of M2-MΦ to distinguish between M2- and M1-MΦ [41]. In PCa, as well as in BPH, the CD163+ MΦ show the typical morphology of MΦ (Figure 4A). In BPH, as well as in PCa, the CD163+ MΦ are mainly localized in the stroma, but are also occasionally found in the glandular lumen (Figure 4A). The density of CD163+ MΦ was significantly (*p* ≤ 0.01) 17.2-fold or 3.8-fold higher in GS7 than in BPH or GS6 (Figure 4B). Interestingly, the density of CD68+ (M1) MΦ was higher than that of CD163+ (M2) MΦ: 24.2- (BPH), 21.1- (GS6), 7.8- (GS7), 5.9- (GS8), or 8.1-fold (GS9) (Figure 3B and Figure 4B).

The density of CD163+ MΦ in the ET regions of the PCa correlated significantly (*p* ≤ 0.04) positively (*r* = 0.45) with the density of GDF-15+ cells in the ET region (Figure 4C,D). However, the density of CD163+ MΦ ET or CD163+ IT did neither correlate with the density of GDF-15+ IT, nor did the density of CD163+ MΦ IT correlate with the density of GDF-15+ ET (Figure 4C,D). These data indicate at least partial co-localization of GDF-15 IR in CD163 IR MΦ.

### 3.4. Preferential Localization of GDF-15 in Transepithelial Migrating CD68+ or CD163+ MΦ and in PGP9.5+ Epithelial Cells in PCa

Using a double immunofluorescence technique, we identified colocalization of GDF-15-IR in CD68+ or CD163+ MΦ (Figure 5 and Figure 6). Interestingly, double immunoreactive GDF-15+/CD68+ were found as transepithelial migrating MΦ, whereas stromal CD68+ MΦ were not GDF-15+. Additionally, PGP9.5+ NE cells, which were occasionally GDF-15+, were observed in the glandular epithelium (Figure 5B). Some epithelial cells of the glandular epithelium were found to be GDF15+ (Figure 6).

### 3.5. Relationship of GDF-15, PD-L1, CD68, and CD163 in Luminal Excrescences in PCa Compared to BPH

Immunohistochemistry of serial sections of PCa (GS6), stained with antibodies directed against GDF-15, PD-L1, CD68, or CD163, exhibit luminal excrescences of epithelial cells with positive IR for GDF-15 and PD-L1, and presence of CD68+ MΦ (Figure 7B,C), but not of CD163+ MΦ (Figure 7D). In contrast, luminal excrescences in BPH were only GDF-15+ (Figure 7E,F). Note that the epithelium of PCa free from excrescences also stains for PD-L1, whereas GDF-15 is absent (Figure 7A,B). The epithelium of BPH without excrescences neither stains for GDF-15 nor PD-L1.

### 3.6. Decreased Density of CD4+ Lymphocytes in PCa Compared with BPH and Close Contact of CD19+ B Lymphocytes with Nerves in PCa but Not in BPH

Therefore, to decipher whether there is a relationship between infiltrated T/B lymphocytes and malignancy of PCa, we measured the density of T/B lymphocytes and carried out double-immunofluorescence. CD4+ (T helper cells) were found in ET and IT areas of all GS tested, as well as in BPH (not shown). The density of CD4+ lymphocytes was significantly (*p* ≤ 0.05) 50.5-fold lower in GS6 and significantly (*p* ≤ 0.05) 101.0-fold lower in GS9 than in BPH (Figure 8A). No significant differences concerning densities of CD8+ cytotoxic T cells were found when comparing GS and BPH (Figure 8B). The density of CD19+ lymphocytes was 2.4- (GS6), 1.3- (GS7), 1.9- (GS8), and 1.9-fold (GS9)—however, insignificantly—higher than in BPH (Figure 8C). In addition, we analyzed the potential communication between lymphocytes and PGP9.5+ nerve fibers (Figure 8D). Clusters of CD19+ B lymphocytes were located in direct contact with PGP9.5+ nerve fiber bundles (Figure 8D). The contacts occurred in the stroma of both ET and IT areas in all stages of PCa. No contacts were found between CD19+ and PGP9.5+ cells in BPH. Contacts between nerves and CD4+- or CD8+ T lymphocytes were found in neither PCa nor BPH.

## 4. Discussion

The TME is a hallmark of cancer and plays a major role in cancer pathophysiology [42]. The development of TME is similar to wound healing and repair or regeneration, but proliferation and migration are not self-limited in cancer [43]. Nerves sprout into the TME and create an intratumoral and peritumoral neural microenvironment [36]. Clinical and in vitro investigations showed that nerve fibers are localized around the tumor mass releasing neurotransmitters and neuropeptides with a direct action on cancer cells and modulatory signaling [44]. The nerves are also involved in inflammatory immune response and angiogenesis [45]. Evidence of the stimulatory effect of tumor innervation and nerve signaling in PCa, as well as the potential use of the neurosignaling for diagnosis, prognosis, and treatment, has been recently reviewed [39]. Thus, to investigate the intra- and peritumoral neural microenvironment, we performed IHC staining using PGP9.5, a marker found in neurons but also in a wide variety of NE [46]. Most frequently, PGP9.5+ cells and nerve fibers were localized in the stroma. However, epithelial and NE PGP9.5+ cells were also found within the glandular epithelium [47]. We found in all PCa and BPH samples an extensive network of PGP9.5+ nerve fibers and cells, with the lowest PGP9.5+ areas in BPH, increasing in PCa from GS6 to GS9. These findings confirm and extend the most recently published data showing that PGP9.5+ innervation differed between PCa and BPH [38]. In PCa, controversial effects concerning PGP9.5+ have been described: some authors suggest PGP9.5 is a tumor suppressor, while others attribute to PGP9.5+ potent pro-oncogenic effects that promote tumor growth and contribute to tumor metastasis [48,49]. Regarding cancer progression in the different GS, our findings concerning PGP9.5+ areas in BPH and PCa with increasing GS favor a pro-oncogenic effect of increased innervation, indicated by PGP9.5+ measurements. In this line, it was shown that PGP9.5 is expressed in human prostate cell lines [34]. PGP9.5 is not only found in nerves but also in prostatic NE cells [34]. If released from NE cells in prostatic cancer PGP9.5 may exert paracrine effects to promote the proliferation of neighboring cells [34].

One of the most intriguing cytokines that may play a role in prostate carcinogenesis is GDF-15; GDF-15 has pleiotropic functions at all stages of tumorigenicity and is suggested as a marker to predict aggressive PCa disease [50]. Interestingly, an increase of GDF-15 in serum was associated with injury, inflammation, and malignancy [51]. Here, for the first time, we show a high density of GDF-15+ cells in human PCa biopsies of all GS. In contrast, GDF-15+ cells were nearly absent in BPH. These findings extend previous data showing that the level of GDF-15 mRNA gene expression was significantly associated with higher GS, especially in advanced and more aggressive prostatic tumors [52].

Double immunofluorescence investigations revealed that most GDF-15+ cells were identified as CD68+ or CD163+ MΦ. It is of particular note that some epithelial cells were GDF-15+, with some of them exhibiting morphological features of PGP9.5+ NE cells. NE cells occur in normal prostatic tissue, in PCa, as well as during differentiation of NE tumors and high-grade NE carcinoma [35]. The occurrence of GDF-15 IR in some of these NE PGP9.5+ cells represents a novel finding.

GDF-15 is not only expressed in PCa but can also be found in the prostatic precancerous inflammatory environment [53]. GDF-15 appears to regulate inflammatory pathways in the prostate, with tumor-promoting and/or suppressing functions [54,55]. GDF-15 inhibits the activity of the NF-κB transcription factor, which can be taken to indicate tumor-suppressing properties [56]. Here, for the first time, we show an increase in density of GDF-15+ cells, as well as CD68+ M1- and CD163+ M2-MΦ in PCa, especially in high-grade GS samples. The densities of CD68+ and GDF15+ cells in IT regions were insignificantly higher than in ET regions. However, the positive correlation between the density of CD68+ MΦ and GDF-15+ cells in ET areas and the presence in the luminal excrescences might be a sign to consider the use of GDF-15 as a potential biomarker.

We provide novel data showing the localization of GDF-15-IR in CD68+ M1 or CD163+ M2 MΦ with a significant positive correlation between the density of GDF-15+ cells and the densities of CD68+ MΦ- or CD163+ M2-MΦ. These novel data suggest GDF-15 as a potential prognostic biomarker to estimate the severity of cancer in prostate biopsies and to differentiate PCa from BPH. Our results concerning density and characterization of GDF-15+ MΦ extend data of others showing that GDF-15/MIC-1 mRNA expression—detected by in situ hybridization—only exists in carcinoma, but not in benign tissue [57]. Recently an association between low M2-MΦ density in benign prostate biopsies and a high M2-MΦ density in the PCa, with an increased risk of PCa, has been reported [50]. Here for the first time, we demonstrate colocalization of GDF-15/MIC-1 in M2-MΦ. Understanding the cell-specific expression patterns of GDF-15 provides new insights into its biological functions. Thus, it helps to delineate the physiological and pathological roles of GDF-15 in the human prostate, especially in PCa.

In our study, we provide new findings on the neuronal and T/B cell signatures in PCa, as compared to BPH, and demonstrate specific neuroimmune connections. Apparently, GDF-15 IR was not localized in T and B lymphocytes. The relevance of these findings becomes evident in the following discussion. Increasing evidence indicates that CD19+ B cells support tumor growth, whereas the lack of mature B cells decreases tumor progression [58]. Connections between the nervous and immune systems have been extensively described, as well as their regulation, e.g., B cells express a higher density of adrenergic receptors (β-AR) than CD4+ T cells, and their stimulation elevates the intracellular concentrations of cAMP [59]. Nerves are currently considered regulators of cancer initiation, progression, and metastasis. In this context, we found that CD19+ B lymphocytes were located in the stroma of PCa in close proximity or in direct contact with PGP9.5+ nerves, but not in BPH. It is reasonable to suggest that B cells present in the vicinity of nerves, as shown in this study, may have an inhibitory function on their protumorigenic properties. Thus, solid tumors in humans often contain significant B cell populations, suggesting a role for these cells in cooperating with other resident cells to influence the tumor microenvironment [60].

We found PD-L1+ in luminal excrescences of the prostate glands that were additionally GDF-15+ and sometimes being identified as CD68+ MΦ, whereas in BPH the excrescences were solely GDF15+, without being PD-L1+ or CD68+. PD-L1 is considered an immune checkpoint as a key regulator of the threshold of immune response and peripheral immune tolerance [61]. On tumor cells, the interaction between programmed death 1 receptor (PD-1, PDCD1) and its ligand programmed death-ligand 1 (PD-L1, CD274) is a critical immunosuppressive mechanism in cancer [62]. In particular, the preferential localization of GDF-15 in excrescences may serve to support future research on the validity of markers such as PD-L1, GDF-15, and CD68 in urine or seminal plasma, for the early and non-invasive detection of PCa, or to define the degree of severity, or be utilized as a non-invasive biomarker of PCa as compared to BPH. For discriminating between PCa and BPH in the “PSA gray zone”, the occurrence of PD-L1, GDF-15, and CD68 immunoreactive cells may also be used as an additional marker similar to urinary molecular PCa risk score (UMPCaRS) by using the sum of three upregulated genes (PDLIM5, GDF-15, THBS4) [63]. The significance of our observation indicating colocalization of GDF-15 and the presence of PD-L1 in luminal excrescences remains to be shown. It is noteworthy to mention that we found luminal excrescences in all stages of PCa. However, with differences in the colocalization pattern of GDF-15 and PDL1. Not all excrescences were GDF-15+ and PDL1+, and some were only PDL1+ or GDF15+ or negative for both markers. However, in BPH, GDF-15+ and PDL1+ cells were never found in excrescences. From GS6 up to GS9, GDF15+ and PD-L1+ excrescences were found and markedly increased in GS9. As PD-L1 is a known marker of tumor progression [64,65], our findings of the colocalization of PDL1 with GDF-15 in PCa but not in BPH is a novel discovery that could be used in future, prospective studies as biomarkers of malignancy in tissue biopsies or cells/RNA in urine or seminal plasma [66,67].

## 5. Conclusions

In summary, the current study indicates that density of innervation, CD68+ (M1), CD163+(M2) MΦ, and GDF15+ cells partly increase in PCa according to the degree of malignancy. GDF-15+ cells are identified mainly as CD68+, or CD163+ MΦ, but sometimes also as PGP9.5+ NE cells. GDF-15+ cells in epithelial excrescences of PCa suggests that GDF-15 can be found in seminal plasma and/or urine. Thus, GDF-15 in seminal plasma and/or urine could be utilized as a non-invasive biomarker of PCa as compared to BPH. The co-presence of PD-L1+ and GDF-15+ excrescences is proposed as a new parameter beyond PSA to define tumor grade progression in prostate biopsies in addition to other common (immuno)histological markers of PCa. Nevertheless, future/prospective clinical studies are necessary to test our hypotheses by using an increased number of patients.

## Figures and Tables

**Figure 1 cancers-14-04591-f001:**
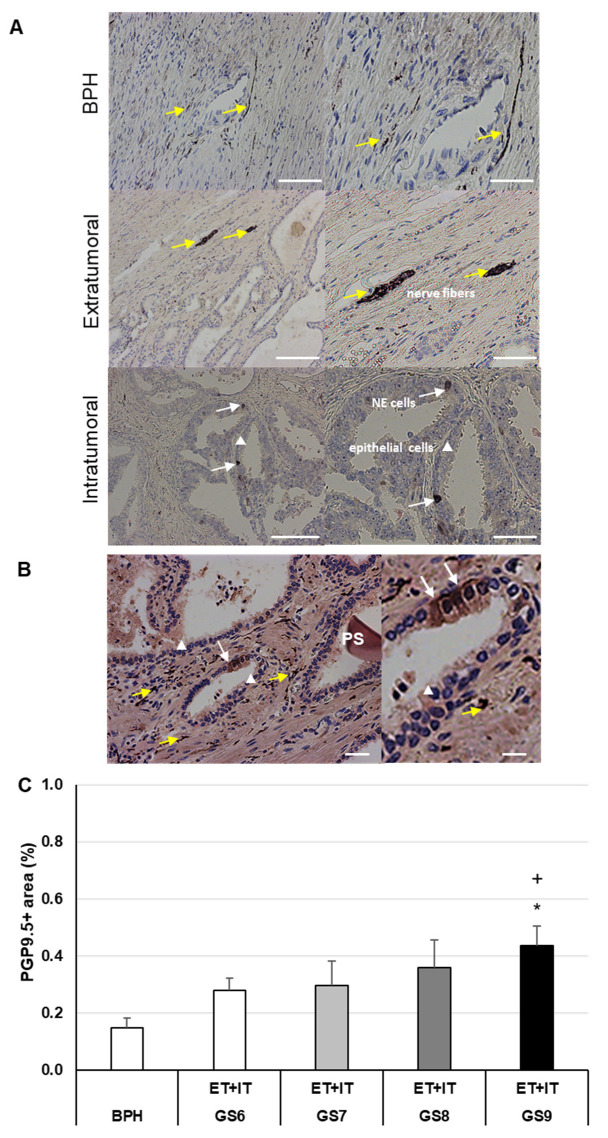
PGP9.5 (~UCHL1) IHC of human BPH and PCa. (**A**) Representative images of BPH and extratumoral [ET] or intratumoral [IT] PCa regions. (**B**) PGP9.5+ epithelial cells. Nerve fibers are marked with yellow arrows, PGP9.5+ neuroendocrine cells (NE) with white arrows, epithelial lining cells are indicated with a white triangle, and a prostate stone (PS) is also seen. (**C**) Percentage of PGP9.5+ area in BPH and PCa with different Gleason scores (GS). The data show means + SEM; significance: * *p* ≤ 0.05 vs. BPH; + *p* ≤ 0.05 vs. GS6. Scale bar from left to right (**A**) 200 and 100 µm; (**B**) 100 and 50 µm.

**Figure 2 cancers-14-04591-f002:**
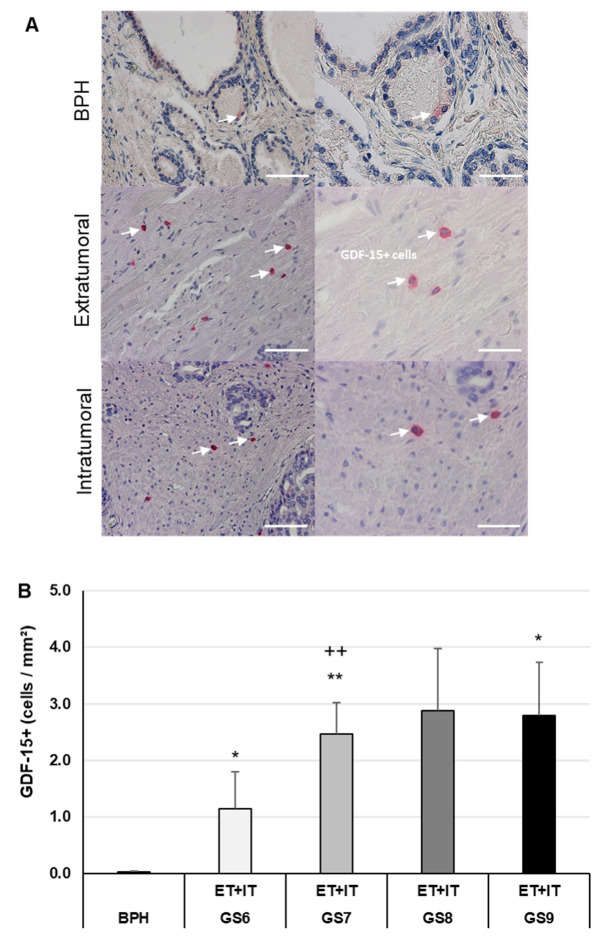
GDF-15 IHC of human BPH and PCa. (**A**) Representative images of BPH and extratumoral [ET] or intratumoral [IT] PCa regions. (**B**) Density of GDF-15+ cells in BPH and PCa (ET; IT) with different Gleason scores (GS). GDF-15+ cells are marked with white arrows. The data show means + SEM; significance: * *p* ≤ 0.05, ** *p* ≤ 0.01 vs. BPH; ++ *p* ≤ 0.01 vs. GS6. Scale bars: left column 200 µm, right column 100 µm.

**Figure 3 cancers-14-04591-f003:**
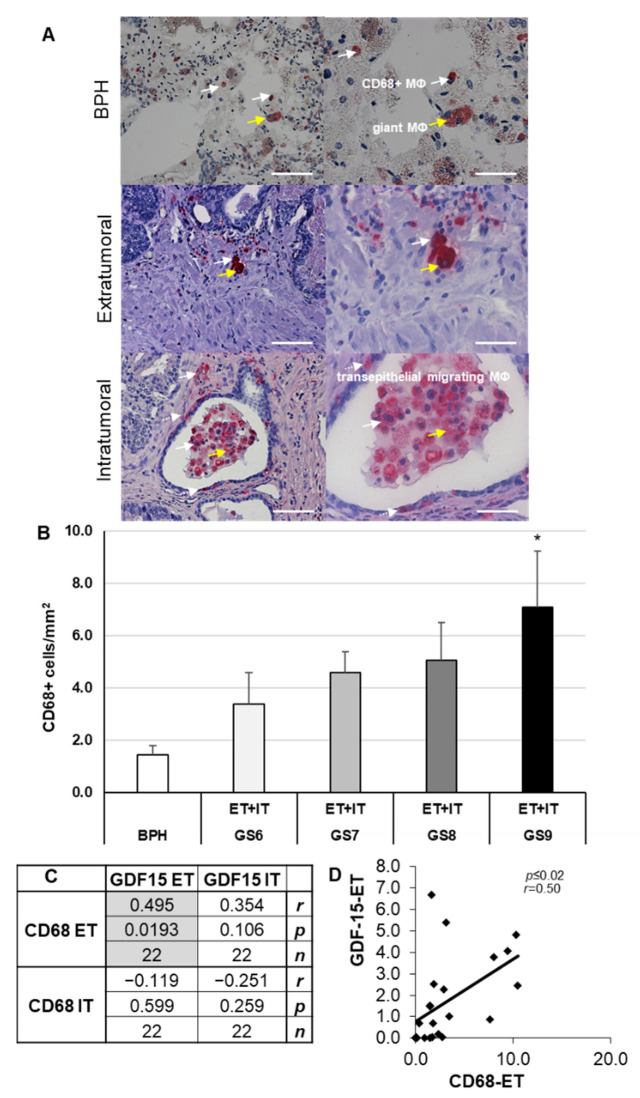
CD68+ MΦ IHC of human BPH and PCa. (**A**) Representative images of CD68+ MΦ in BPH and extratumoral [ET] or intratumoral [IT] PCa regions; CD68+ MΦ are marked with white arrows; multinucleated MΦ are marked with yellow arrows, and transepithelial migrating MΦ are indicated with white dotted arrows. (**B**) Density of CD68+ MΦ in BPH and PCa with different Gleason scores (GS). (**C**,**D**) Pearson’s correlation coefficient, significant correlation was marked with gray. The data show means + SEM; significance: * *p* ≤ 0.05 vs. BPH. Scale bars: left column 200 µm, right column 100 µm. Pearson product moment correlation between ET and IT of CD68+ and GDF-15+ (**C**,**D**). Correlation coefficient (*r*), *p*-value (*p*) and the number of samples (*n*).

**Figure 4 cancers-14-04591-f004:**
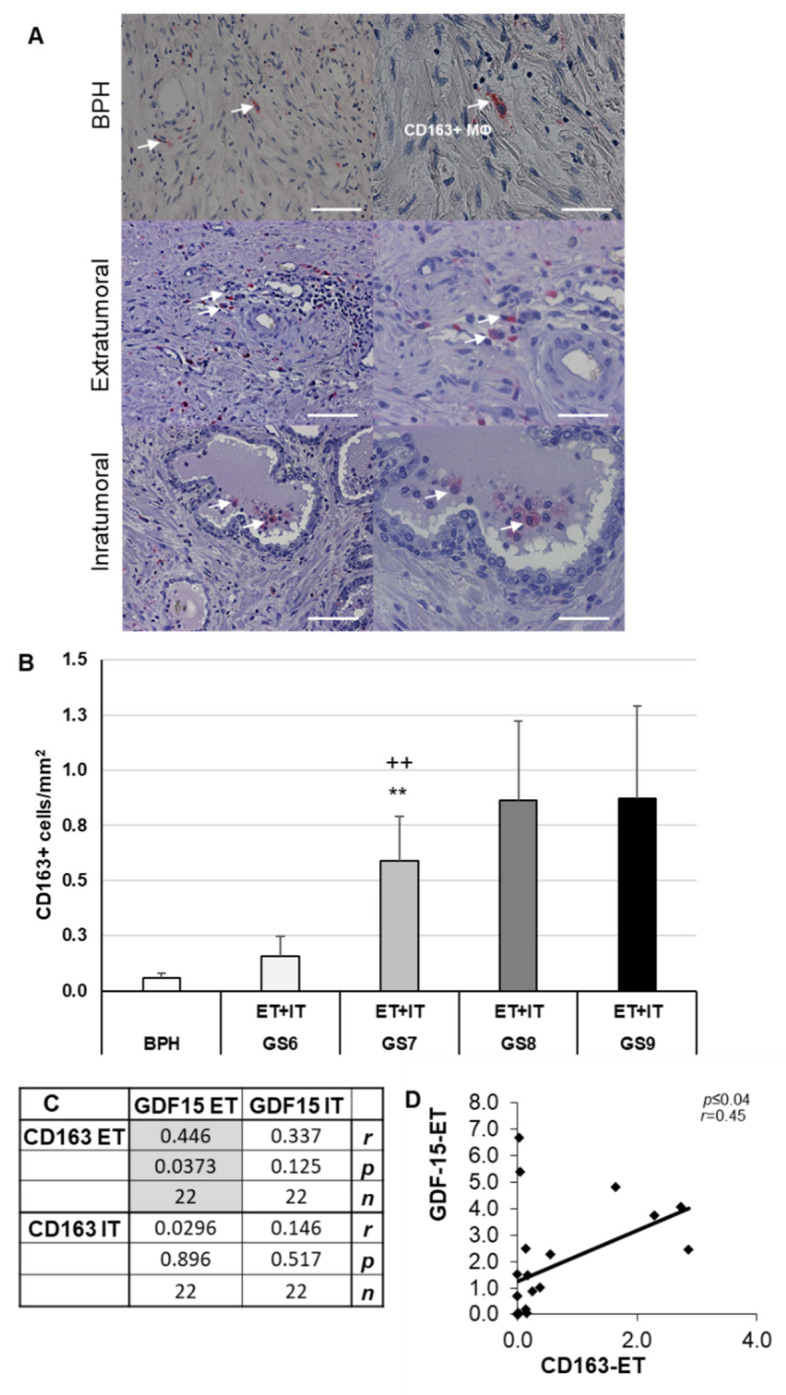
CD163+ MΦ IHC of human BPH and PCa. (**A**) Representative images of BPH and extratumoral [ET] or intratumoral [IT] PCa regions of CD163+ cells; CD163+ MΦ are marked with white arrows. (**B**) Density of CD163+ MΦ in BPH and PCa with different Gleason scores (GS). (**C**,**D**) Pearson’s correlation coefficient, a significant correlation was marked with gray. The data show means + SEM; significance: ** *p* ≤ 0.01 vs. BPH; ++ *p* ≤ 0.01 vs. GS6. Scale bars: left column 200 µm, right column 100 µm. Pearson product moment correlation between ET and IT of CD163+ and GDF-15+ (**C**,**D**). Correlation coefficient (*r*), *p*-value (*p*) and the number of samples (*n*).

**Figure 5 cancers-14-04591-f005:**
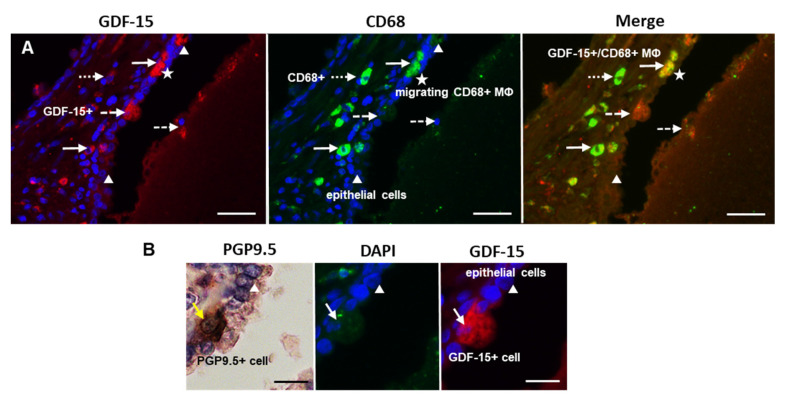
(**A**) Double-immunofluorescence of GDF-15 (Cy3, red) and CD68 (Alexa Fluor^®^ 488, green) in PCa. White arrows indicate the double-positive (GDF-15+/CD68+ MΦ); white arrows with a dotted line indicate single-CD68+ MΦ; and arrows with a dashed line mark single-GDF-15+; white stars indicate transepithelial migrating CD68+ MΦ; white triangles mark epithelial lining cells of the prostate glands; scale bar 100 µm. (**B**) Photos of cross sections showing neuroendocrine PGP9.5+ and GDF-15+ cells; PGP9.5+ cells are marked with yellow arrows and GDF-15+ cells with white arrows; white triangles mark epithelial lining cells; nuclei were counterstained with DAPI (blue); scale bar 25 µm.

**Figure 6 cancers-14-04591-f006:**
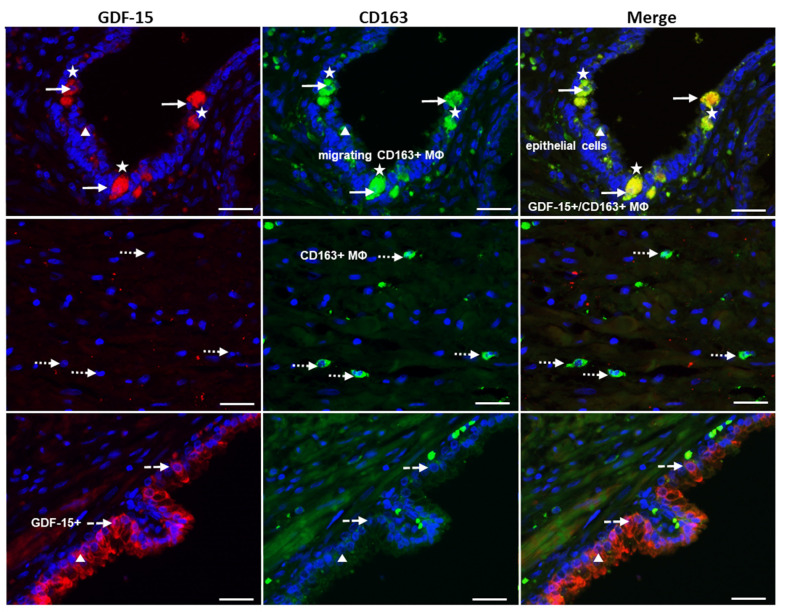
Double-immunofluorescence of PCa using GDF-15 (Cy3, red) and CD163 (Alexa Fluor^®^ 488, green) antibodies. White arrows indicate double-positive (GDF-15+/CD163+ MΦ); white arrows with a dotted line mark single-CD163+ MΦ; and arrows with a dashed line indicate single-GDF-15+; white stars mark transepithelial migrating CD163+ MΦ; white triangles indicate epithelial lining cells of the prostate glands. Nuclei were counterstained with DAPI (blue); scale bar 100 µm.

**Figure 7 cancers-14-04591-f007:**
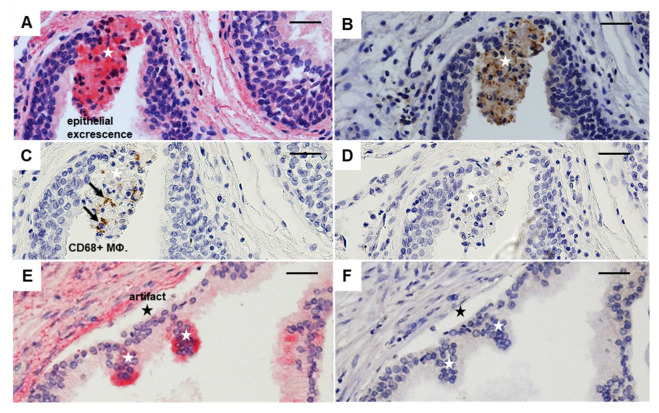
Epithelial cells with luminal excrescences in the prostatic gland. Representative immunohistochemistry of serial sections of GS6 PCa (**A**–**D**) or BPH (**E**,**F**), stained with GDF-15 (**A**,**E**), PD-L1 (**B**,**F**), CD68 (**C**), CD163 (**D**). White stars mark epithelial excrescence and black stars an artifact; black arrows indicate CD68+ MΦ. Nuclei were counterstained with hematoxylin; scale bar 100 µm.

**Figure 8 cancers-14-04591-f008:**
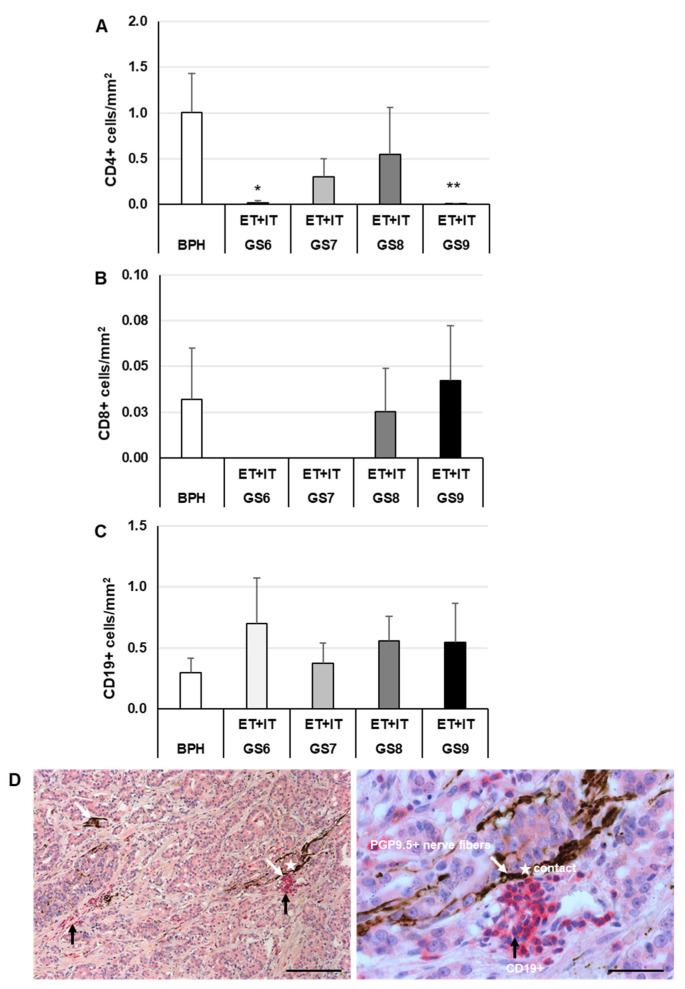
IHC of lymphocytes and nerve fibers in human BPH and extratumoral [ET] or intratumoral [IT] PCa regions. (**A**) Density of CD4+, T helper cells; (**B**) of CD8+, cytotoxic T cells; and (**C**) density of CD19+, B lymphocytes. The data show means + SEM; significance: * *p* ≤ 0.05, ** *p* ≤ 0.01, vs. BPH. (D) Representative photos of the contact (white star) between B lymphocytes (CD19+) and PGP9.5+ nerve fibers in PCa. The white arrows mark PGP9.5+ nerve fibers and black arrows mark clusters of B lymphocytes (CD19+). Nuclei were stained with hematoxylin. Scale bar from left to right 200 and 100 µm.

## Data Availability

Not applicable.

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
